# Establishment of an Intradermal Canine IL-31-Induced Pruritus Model to Evaluate Therapeutic Candidates in Atopic Dermatitis

**DOI:** 10.3390/vetsci10050329

**Published:** 2023-05-04

**Authors:** Jason Pearson, Renato Leon, Haley Starr, Sujung Jun Kim, Jonathan E. Fogle, Frane Banovic

**Affiliations:** 1College of Veterinary Medicine, University of Georgia, Athens, GA 30602, USA; jason.pearson26@uga.edu (J.P.); rgl84554@uga.edu (R.L.); haley.starr@uga.edu (H.S.); 2Boehringer Ingelheim Animal Health, Athens, GA 30601, USA; sujung.kim@boehringer-ingelheim.com (S.J.K.); jonathan.fogle@boehringer-ingelheim.com (J.E.F.)

**Keywords:** canine, pruritus, interleukin-31, atopic dermatitis, oclacitinib

## Abstract

**Simple Summary:**

Experimental research using canine itch models advanced the development of novel medications that inhibit different itch-induced pathways. This study aimed to investigate the immediate/delayed pruritus responses/behaviors observed after the intradermal administration of recombinant canine interleukin-31 (IL-31) to healthy dogs (intradermal IL-31-induced pruritic model) and the reversal of the induced pruritic behaviors in the same dogs using oral oclacitinib (JAK inhibitor). The itch behaviors were video-recorded for 300 consecutive min, and two blinded investigators reviewed all the video recordings. Intradermal IL-31 administration to healthy dogs caused a significant increase in pruritic behaviors, which were significantly reduced after oral oclacitinib administration. Significant delayed pruritic responses at 150–300 min after IL-31 injections were observed, whereas intradermal IL-31 failed to induce acute itch in the dogs within the first 30 min. Intradermal injection of IL-31 induces delayed itch responses in dogs that are diminished by the effect of oclacitinib, an oral JAK inhibitor.

**Abstract:**

Pruritic models in healthy dogs utilizing intravenous administration of interleukin 31 (IL-31) bypass the “natural” itch sensation in AD, which is initiated by pruriceptive primary afferent neurons in the skin. This study aimed to evaluate the immediate/delayed pruritus responses and the pruritic behaviors observed in an intradermal IL-31-induced pruritic model of healthy dogs and the anti-pruritic effect of oclacitinib on said model. In Phase 1, all the dogs were randomized and video-recorded for 300 min after intradermal canine recombinant IL-31 injections (1.75 µg/kg) and vehicle (phosphate-buffered saline) injections. In Phase 2, all the dogs received oral oclacitinib (0.4–0.6 mg/kg, twice daily for 4 consecutive days and once daily on day 5), with the intradermal IL-31 injection performed on day 5. Two blinded investigators reviewed the pruritic behaviors in all the video recordings. Intradermal IL-31 administration to healthy dogs caused a significant increase in the total (*p* = 0.0052) and local (*p* = 0.0003) seconds of pruritic behavior compared to the vehicle control. Oral oclacitinib administration significantly reduced the total (*p* = 0.0011) and local (*p* = 0.0156) intradermal IL-31-induced pruritic seconds; there was no significant difference in pruritic seconds between the vehicle and oclacitinib within the IL-31 groups. Significant delayed pruritic responses at 150–300 min after IL-31 injections were observed, and intradermal IL-31 failed to induce acute itch (first 30 min). Intradermal injection of IL-31 induces delayed itch responses in dogs that are diminished by the effect of oclacitinib, an oral JAK inhibitor.

## 1. Introduction

Atopic dermatitis (AD) is a chronic, frequently relapsing, inflammatory skin disease affecting humans and dogs with pruritus (itch) as a cardinal symptom [1,2]. Increased pruritus in AD results in significant sleep disturbances and poor quality of life for human and canine AD patients, as well as pet owners [1,2,3,4]. In addition, pruritic behaviors damage the skin and lead to the exacerbation of skin barrier dysfunction and allergic inflammation, resulting in a so-called “itch-scratch cycle” [1,2,5].

Interleukin-31 (IL-31), a T helper 2 cytokine involved in the itch sensation, has been a focus of novel treatments for pruritus in human and canine AD [6,7,8]. IL-31 produces signals by activating a receptor complex composed of IL-31 receptor A (IL-31RA) and the oncostatin M receptor (OSMR), with the downstream activation/phosphorylation of several kinase signaling pathways, such as the Janus kinase (JAK) signal transducer and activator of transcription (STAT) and phosphatidylinositol-3 kinase (PI3K/AKT) [6]. It has been shown that IL-31 can directly induce an itch sensation via its heterodimeric receptor complex on free nerve endings and/or through promoting other cell types (e.g., keratinocytes, fibroblasts) to release/secrete additional pruritogens that then evoke the itch signal [9,10,11]. In the last decade, drugs that target the IL-31 pathway, such as lokivetmab (caninized monoclonal antibody against canine IL-31) [12] and nemolizumab (humanized monoclonal antibody against human IL-31RA) [13], have significantly reduced itch and cutaneous inflammation scores in canine and human AD patients, respectively. Therefore, IL-31 is currently regarded as a significant pruritogen in humans and dogs.

The mechanisms underlying pruritus have been evaluated using experimental animal models with the administration of various pruritogens. Experimental research using canine itch models have advanced the development of novel medications that inhibit different itch-induced pathways. The canine IL-31-induced pruritic model was developed utilizing intravenous injections of recombinant canine IL-31 in healthy dogs [14,15]. Subsequently, this intravenous IL-31 canine pruritus model was used to determine the onset and duration of action of the current commonly used treatments for canine AD such as oclacitinib (Apoquel, Zoetis; Kalamazoo, MI, USA), lokivetmab (Cytopoint, Zoetis) and glucocorticoids [15,16,17]. Interestingly, a detectable value of canine IL-31 in the serum was found in only 57% (127 of 223) of dogs with AD [14]. Furthermore, the “natural” itch sensation pathway in AD is caused by the activation of pruriceptive primary afferent receptors in the skin (“periphery”), with the ascending signal passing through dorsal root ganglion (DRG) neurons and spinal cord circuits. The DRG synapses in the dorsal horn of the spinal cord further induce the activation of ascending itch-signaling pathways to the central nervous system [18,19,20]. Therefore, experimental research regarding skin-disease-associated pruritus is commonly performed using an epicutaneous application (e.g., dust mite extracts) or intradermal injections of pruritogens across species, which contrasts with the previously published intravenous canine IL-31-induced pruritic model.

The objectives of the present study were to characterize intradermal canine recombinant IL-31-provoked pruritic behaviors in 10 healthy laboratory dogs and to evaluate the anti-pruritic effect of oclacitinib (Apoquel) on this intradermal IL-31-induced canine pruritic model.

## 2. Materials and Methods

### 2.1. Study Population

This prospective, randomized, controlled crossover study included 10 healthy research beagle dogs. Every dog in the study received all three interventions (vehicle, IL-31 administration, IL-31 administration with oclacitinib). This number of dogs was deemed sufficient for this experiment in order to have a 90% power to detect a significant 2-fold difference in the pruritic scores (i.e., seconds) between treatment groups (controls and IL-31 intervention) (*p* = 0.05 with SD of 30%). All dogs were male-castrated, weighed 8.5 kg to 13.0 kg, aged from 1 to 6 years and were housed in their kennels under standard university conditions. To avoid any potential influence of other medications in this study, there were no drugs provided to any dogs for at least 12 weeks before enrollment. Before and during the study enrollment, the investigators performed a physical examination and ruled out dermatologic and systemic illnesses. All the experiments herein were approved by the Institutional Animal Care and Use Committee (IACUC; number A2020 05-016-Y3-A9).

### 2.2. Interventions and Protocol

Recombinant canine IL-31 was designed and produced as previously described [14]. Briefly, the mammalian HEK293 expression system was used to produce canine IL-31, whereas mass spectrometry (tryptic digest and mapping) and N-terminal sequencing were utilized to confirm the canine IL-31 sequence. The biological activity of canine IL-31 was determined using IL-31-induced STAT3 phosphorylation assay in canine DH82 cells.

The study was performed in two phases (Phase 1 and 2). A 4-week wash-out period between all interventions was determined based on a previous IL-31-induced pruritus study of dogs. The skin areas for the IL-31 intradermal injections were clipped on the right lateral thoracic region ≥ 72 h before each injection. This clipping was performed to avoid any potential occurrence of localized skin irritation in the clipped areas where the recombinant IL-31/saline would be injected for the evaluation of pruritic behaviors.

In Phase 1, all the dogs were randomized using statistical computer software (Prism 9.0; GraphPad Software; La Jolla, CA, USA) to receive intradermal injections of either recombinant canine IL-31 diluted in sterile phosphate-buffered saline (PBS) at 1.75 μg/kg (IL-31 group) [14,15,16,17] or sterile PBS (vehicle group, negative controls) in the right lateral thoracic regions. All the dogs, after the wash-out period, received the alternate intradermal injection (IL-31 or PBS).

In Phase 2, all the dogs received oral oclacitinib (Apoquel, Zoetis Inc., New York, NY, USA) at 0.4 to 0.6 mg/kg twice daily for four consecutive days and only in the morning on day 5 (total of 9 oral oclacitinib administrations per dog). Intradermal injections of recombinant canine IL-31 diluted in sterile phosphate-buffered saline (PBS) at 1.75 μg/kg were given to all the dogs receiving oclacitinib on day 5. Recombinant canine IL-31 was injected into the lateral thorax 1 h post oral oclacitinib administration as previously described in a published evaluation of oclacitinib’s effects on the intravenous IL-31-induced pruritic model [17].

### 2.3. Video Recordings

To mimic the daily natural environment and avoid any potential effects on the dogs’ behaviors, the dogs were always recorded in their kennels. Camera equipment for the video recordings was placed above the kennels at least one week before the interventions (e.g., saline, IL-31) so that the dogs could be acclimated to the recordings. In addition, the dogs were always recorded in their daily runs at the same time of day to avoid any deviation between treatment groups. High-definition GoPro Hero Session (GoPro, Inc.; San Mateo, CA, USA) cameras were used for every recording. To avoid direct interactions between the dogs during the recordings, each dog was housed as a single dog in its kennel during the study. Before each recording, the cameras were placed above the kennels, and we checked that the cameras covered the whole kennel space and that each dog was fully visible for the evaluation during the recording period. The video recordings were performed without the presence of the investigators during the filming, as the investigator’s presence can affect the dogs’ behaviors.

All video recordings were performed for a total time of 300 min (5 h) post-administration of saline and recombinant IL-31. A duration of 300 min for the IL-31/vehicle video recordings in this study was chosen based on the published studies of intravenous IL-31-induced itch in dogs [15,16,17].

### 2.4. Assessment of Pruritus

Pruritic behavior measurements were evaluated using the canine ethogram for itch behavior (Appendix A), which assesses general and local (e.g., areas of intradermal injections) itch behaviors [21,22]. The investigators evaluated selected itch behaviors of licking, biting/chewing, head shaking, scratching, rolling, scooting, pawing, and rubbing [21,22]. In contrast, pain behavior was considered when there was vocalization after treatment injections and if the dogs showed flinching at the site of treatment injection [21,22].

The interobserver reliability was assessed using three video recordings and revealed a significant correlation in the pruritic scores for identical videos (Spearman’s r = 0.94; *p* < 0.0001) between the two investigators. The video recordings were randomized and divided between the two trained investigators. The video recordings were coded to prevent (blind) the investigators from knowing the types of substances administered in the videos. For every video recording, each investigator evaluated the duration of each pruritic episode in seconds and the type of pruritic behavior.

The initial interobserver reliability evaluations of IL-31-induced pruritus in the dogs showed numerous consecutive pruritic behaviors (from paw licking to head shaking to scratching the trunk to licking paws, etc.). The video evaluators initially tried to quantify each type of pruritic behavior (e.g., licking, chewing, head shaking) in total seconds; however, repeated video evaluations in slow motion needed to be performed, which resulted in an extensive time commitment for every video evaluation. To simplify the assessment of the types of pruritic behaviors and render it time-efficient for future study evaluations, sleeping and observed pruritic behaviors, such as licking, biting/chewing, head shaking, scratching, rolling, scooting, pawing, and rubbing, were measured using a previously reported categorical scoring system of ‘Yes/No’ (i.e., 1 or 0) at 1 min intervals.

### 2.5. Statistical Analysis

All analyses were performed using GraphPad Prism v.9.0. The total (during 5 h) pruritic seconds (generalized and local), local pruritic seconds and pruritic behaviors (e.g., licking, chewing, head shaking) were compared between the vehicle, IL-31 and IL-31 with oclacitinib groups. An initial normality assessment of data was performed with a D’Agostino and Pearson omnibus normality test. Depending on the normality test outcome, the data were analyzed using the parametric repeated-measures ANOVA or the nonparametric Friedman test. Acute (first 30 min) and delayed (after 30 min) itch effects of recombinant canine IL-31 on healthy dogs were investigated using a two-way mixed effects model for every 30 min time period (0–30min, 30–60min, 60–90min, 90–120 min, 120–150 min, 150–180 min, 180–210 min, 210–240 min, 240–270 min, 270–300 min) with the Holm–Šídák correction test for multiple comparisons. Statistical significance was set at *p* < 0.05.

## 3. Results

None of the dogs showed any pain-related behaviors (e.g., flinching, vocalization) during any intervention. In addition, no adverse events beyond induced pruritic behaviors were observed after intradermal IL-31 administration in any of the dogs, and none of the dogs developed erosions or ulcerations from the pruritic behaviors.

### 3.1. Pruritus Scores during 5 h Interval

Canine recombinant IL-31 intradermal administration to purpose-bred healthy beagle dogs caused a significant increase in the total (Figure 1a,b; generalized and local; IL-31, median = 2117; *p* = 0.0052) and local (median = 115; *p* = 0.0003) seconds of pruritic behavior compared to the vehicle group (median total pruritic seconds = 188; median local pruritic seconds = 2). Oral oclacitinib administration (median 0.54 mg/kg) significantly reduced the total (median = 187; *p* = 0.0011) and local (median = 26; *p* = 0.0156) intradermal IL-31-induced pruritic seconds compared to the IL-31-only group. There was no significant difference in the total or local pruritic seconds between the vehicle and oclacitinib with IL-31 groups (*p* = 0.79 for total pruritic seconds; *p* > 0.99 for local pruritic seconds).

### 3.2. Acute and Delayed Pruritus Behaviors after Intradermal IL-31 Administration

To determine any acute (first 30 min) and delayed (after 30 min to 300 min) IL-31-induced itch behaviors, we evaluated the total and local pruritic behaviors in seconds for every 30 min interval during the 300 min video observations (Figure 2a,b; 0–30 min, 30–60 min, 60–90 min, 90–120 min, 120–150 min, 150–180 min, 180–210 min, 210–240 min, 240–270 min, 270–300 min).

There was no significant difference in pruritic seconds observed in the first 30 min (acute itch) between the IL-31 (total itch, median = 143; local itch, median = 193) and vehicle (total itch, median = 143; local itch, median = 193) groups.

The observations for delayed IL-31-induced pruritic effects revealed the most significant increases (Figure 2a; *p* < 0.001) in the total pruritic seconds after IL-31 administration to dogs at the following time intervals: 120–150 min (IL-31, median = 186 s; vehicle, median = 13 s); 150–180 min (IL-31, median = 201 s; vehicle, median = 14 s); 180–210 min (IL-31, median = 291 s; vehicle, median = 15 s); 210–240 min (IL-31, median = 235 s; vehicle, median = 19 s) and 240–270 min (IL-31, median = 484 s; vehicle, median = 8 s).

For the local itch behaviors (Figure 2b), there were only significant increases in pruritic seconds for the following time intervals: 180–210 min (*p* = 0.0005; IL-31, median = 18 s; vehicle, median = 0 s) and 240–270 min (*p* = 0.0021; IL-31, median = 27 s; vehicle, median = 0 s).

### 3.3. Pruritic Behaviors during 5 h Interval

Intradermal IL-31 administration, compared to the placebo vehicle group, induced consistent generalized pruritic behaviors in the healthy dogs, with scratching, biting/chewing, licking, and head shaking as the most commonly noted behaviors.

The type of upregulation during intradermal IL-31 administration induced a significant increase in categorial minute (Yes/No) pruritic behaviors of scratching (Figure 3a; *p* = 0.0004), chewing/biting (Figure 3b; *p* = 0.0001), licking (Figure 3c; *p* ≤ 0.0001) and head shaking (Figure 3d; *p* = 0.0008) compared to the vehicle group during the observation period of 300 min. Treatment with oclacitinib significantly reduced all IL-31-induced pruritic behaviors including scratching (Figure 3a; *p* = 0.0004), biting/chewing (Figure 3b; *p* = 0.0001), licking (Figure 3c; *p* ≤ 0.0001) and head shaking (Figure 3d; *p* = 0.0008). There was no statistically significant difference in pruritic behaviors between the vehicle and oclacitinib with IL-31 groups (Figure 3a,b).

## 4. Discussion

In this study, the intradermal administration of IL-31 was shown to elicit statistically significant pruritic behaviors in healthy dogs over 5 h compared to the vehicle control group without causing apparent severe cutaneous self-injury (e.g., erosions/ulcers). In contrast to previous canine intravenous IL-31-induced pruritic studies [15,16,17], the present study’s design included a crossover for each dog to ensure that it was exposed to all interventions. This resulted in a minimal effect of individual variability on pruritic behaviors between dogs. In addition, oclacitinib, an oral JAK inhibitor registered for the treatment of canine allergic dermatitis, successfully reduced the intradermal IL-31-induced pruritic behaviors in this study. There was no significant difference observed in the pruritic scores between the vehicle control and oclacitinib with IL-31 groups. These results regarding the efficacy of oclacitinib in attenuating intradermal IL-31 signaling in dogs are in agreement with the previous oclacitinib studies on the intravenous IL-31-induced pruritic canine model [15,16].

The administration of IL-31 to healthy humans, mice, rats, cynomolgus monkeys and dogs induces the upregulation of pruritic behaviors [14,23,24,25]. Intravenous and intradermal IL-31 administration to cynomolgus monkeys induced immediate (e.g., in the first hour) and delayed itch over a 3 h observation period [24], whereas IL-31 injections of healthy humans (prick testing) [25] and mice (intradermal) [23] showed a slower onset, with a delayed pruritic effect after 2 to 3 h post-injection. Based on these findings [23,24,25], it was concluded that IL-31 does not induce an immediate but rather a delayed itch response in humans and mice, which likely occurs as a result of its effects on IL-31R-expressing keratinocytes, nerves and/or immune cells with the subsequent release of secondary mediators and pruritogens. In the previous canine IL-31 pruritus studies [14,15,16,17], there were no assessments of the IL-31 effect on immediate and delayed itch responses in healthy dogs. The results of our study showed no significant immediate (first 30 min) increase in the total or local pruritic seconds after intradermal canine IL-31 administration; however, a delayed itch response with the most significant increases in total pruritic behaviors was observed between 150–300 min post-IL-31-injection (Figure 2). These results indicate that intradermal IL-31 delivery in dogs may resemble the delayed itch responses observed in humans and rodents (slower onset) in contrast to cynomolgus monkeys (rapid onset). Further studies evaluating canine IL-31 signaling in neurons are needed to better understand delayed IL-31-induced itch in dogs.

Similar to studies on cynomolgus monkeys [24], the intradermal administration of IL-31 caused generalized pruritic behaviors that were not limited to the intradermal injection site area but occurred at distant sites. A plausible explanation could be that canine IL-31, after intradermal administration, enters the systemic circulation and induces itch by directly targeting spinal cord circuits, which has been observed in rodents after the intrathecal administration of IL-31 [9]. Interestingly, intradermal injections of substance P, a neurokinin-1 receptor agonist [26], induced vomiting in healthy dogs (10 out of 20 dogs tested), confirming the idea that some of the intradermally injected compounds can be systemically absorbed by dogs [27].

In the previous intravenous canine IL-31 pruritus model studies [14,15,16,17], itch behaviors of licking/chewing of the paws, flanks and/or anal region, scratching of the flanks or neck, floor pawing, head shaking and scooting were evaluated; however, there were no specific descriptions of which types of pruritic behaviors dominated after intravenous IL-31 delivery. Different pruritic behaviors were evaluated in this study using the published and well-defined canine pruritic ethogram, which was developed by collecting data from previous itch studies on dogs and used in several canine itch induction studies [21,22]. Among the different pruritic behaviors, scratching, head shaking, and biting/chewing were most commonly recorded when utilizing the categorical “Yes/No” per 1 min interval scoring after the intradermal injection of canine IL-31 in our study. Our initial intention was to assess every itch behavior in our IL-31 study and quantify the duration in seconds. The initial investigations of the video recordings showed that an analysis of all the pruritic behaviors in total seconds can be easily performed; however, the analysis of consecutive changes in different pruritic behaviors in seconds utilizing the slow-forward function for video recordings increased the duration of video evaluation to more than 10 h per dog. Although our categorical “Yes/No” per 1 min scoring system for different pruritic behaviors may contain a bias and is a limitation of our study, the authors felt that the canine IL-31 pruritic model evaluations needed to be performed in a time-efficient way.

## 5. Conclusions

In conclusion, to the best of the authors’ knowledge, this study was the first characterization of the canine intradermal IL-31 pruritic model. Interestingly, injections of canine recombinant IL-31 induce a delayed rather than an acute (first 30 min) itch response, which has been observed in humans and mice. There is a significant medical need for safe and effective treatments for pruritic skin conditions that impair the quality of life of canine patients. In contrast to previous intravenous IL-31-induced canine pruritic studies, the intradermal IL-31-induced pruritic model represents the “natural” itch pathway development of skin diseases from neurons in the skin–dorsal root ganglia–spinal cord–brain. In addition, we used oclacitinib, an oral JAK inhibitor registered for the treatment of canine allergic dermatitis, to successfully ameliorate intradermal IL-31-induced pruritic behaviors in this study and validate the model.

## Figures and Tables

**Figure 1 vetsci-10-00329-f001:**
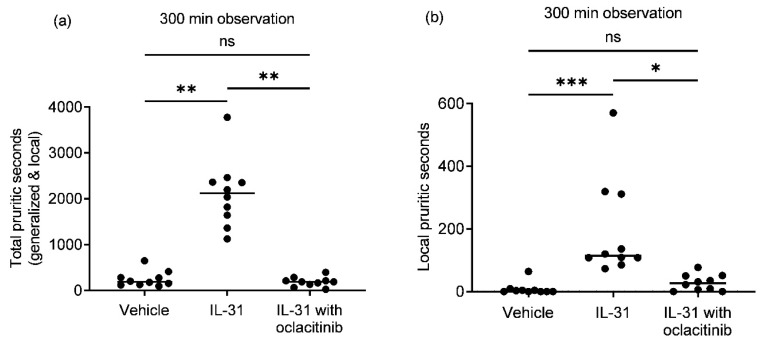
Intradermal canine recombinant interleukin-31 (IL-31) injections induced significant upregulation in total (**a**) and local (**b**) pruritic seconds in 10 healthy dogs compared to the vehicle control groups (Vehicle) during 300 min observations. Oral oclacitinib administration significantly reduced the total and local intradermal IL-31-induced pruritic seconds compared to the IL-31-only group. There was no significant difference in the total or local pruritic seconds between the vehicle and oclacitinib with IL-31 groups. Scatter plot data for each dog’s pruritic score with the median. * *p* ≤ 0.05; ** *p* ≤ 0.01; *** *p* < 0.001.

**Figure 2 vetsci-10-00329-f002:**
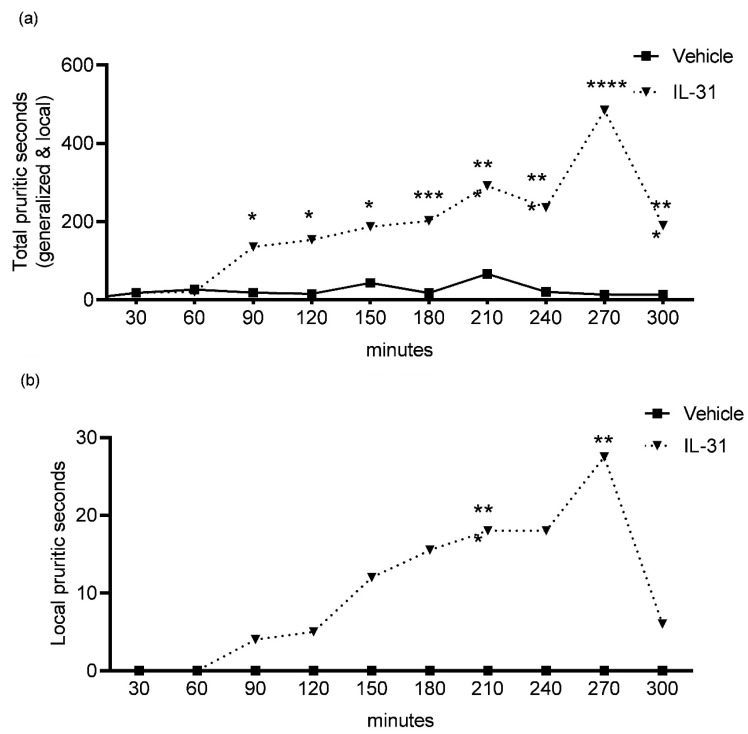
Total (**a**) and local (**b**) pruritic behaviors in seconds for every 30 min interval during 300 min video observations after intradermal canine recombinant interleukin-31 (IL-31) and vehicle (vehicle) injections in 10 healthy dogs. Data are presented as medians. * *p* ≤ 0.05; ** *p* ≤ 0.01; *** *p* < 0.001; **** *p* < 0.0001.

**Figure 3 vetsci-10-00329-f003:**
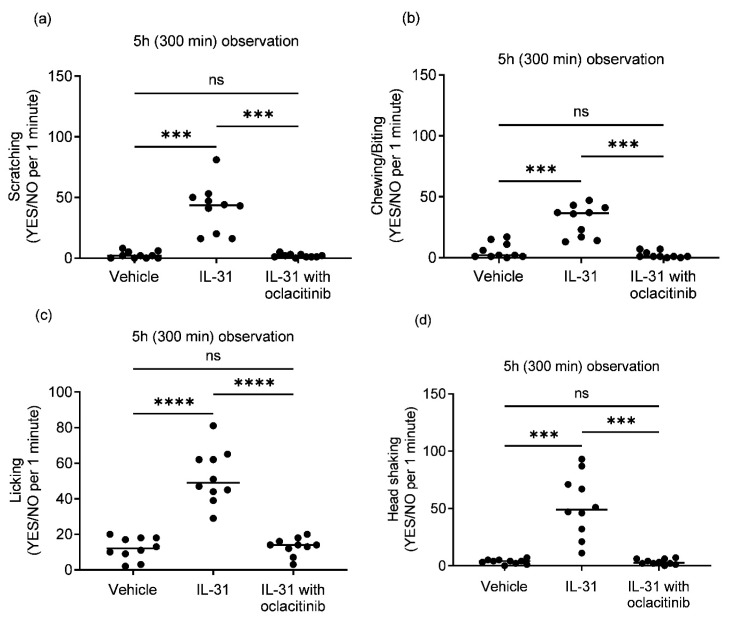
Pruritic behavioral changes in a canine intradermal IL-31 pruritus model (n = 10 dogs) after intradermal injections of canine IL-31 (IL-31), vehicle control (vehicle) and oclacitinib with IL-31 over 300 min intervals after injections. The type of behavior was recorded for each pruritus episode ((**a**) scratching; (**b**) chewing/biting; (**c**) licking; (**d**) head shaking) and scored as “Yes/No” per 1 min interval as a categorical variable. Scatter plot data for each dog with the median are shown. ns; not significant; *** *p* < 0.001; **** *p* < 0.0001.

## Data Availability

Data sharing not applicable. No new data were created or analyzed in this study.

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
