# Peer review of "Establishment of an Intradermal Canine IL-31-Induced Pruritus Model to Evaluate Therapeutic Candidates in Atopic Dermatitis"

_vetsci, 2023, doi:10.3390/vetsci10050329_

Round 1

Reviewer 1 Report

Spelling error "IL31-induced" in the paragraph above Materials and Methods.

Please see my comments in italic: 1. What is the main question addressed by the research?    The authors wanted to determine if intradermal injection of IL-31 could be used as a itch model, and also if administration of oclacitinib will reduce the pruritus. 
2. Do you consider the topic original or relevant in the field? Yes, this topic is original and relevant in this field as allergic dermatitis is the most common disease referred to a dermatologist. Administration of IL-31 intradermally in my opinion, mimics the actual pathogenesis compared to intravenous administration.   Does it address a specific gap in the field? Yes, see my comments above.
3. What does it add to the subject area compared with other published
material? As stated above, the study involved administration of IL-31 directly to the skin, which is the organ that is being tested. 
4. What specific improvements should the authors consider regarding the
methodology? The evaluation of “itch behavior” using ‘yes/no’ could be improved. Understandably, the quantitative evaluation is very time-consuming, but should not be overlooked. Maybe a simplified quantitative evaluation can be considered in the future.     What further controls should be considered? Using different dog breed should be considered. 
5. Are the conclusions consistent with the evidence and arguments presented
and do they address the main question posed? Yes. This study is quite straight forward and the conclusions addressed the main questions
6. Are the references appropriate? Yes.
7. Please include any additional comments on the tables and figures. None.

Author Response

Reviewer 1

Please see my comments in italic:

  1. What is the main question addressed by the research?    

The authors wanted to determine if intradermal injection of IL-31 could be used as a itch model, and also if administration of oclacitinib will reduce the pruritus. 

Answer to the comment: Thank you.

  1. Do you consider the topic original or relevant in the field? 

Yes, this topic is original and relevant in this field as allergic dermatitis is the most common disease referred to a dermatologist. Administration of IL-31 intradermally in my opinion, mimics the actual pathogenesis compared to intravenous administration.  

Answer to the comment: Thank you.

Does it address a specific gap in the field? Yes, see my comments above.

Answer to the comment: Thank you.

  1. What does it add to the subject area compared with other published
    material? 

As stated above, the study involved administration of IL-31 directly to the skin, which is the organ that is being tested. 

Answer to the comment: Thank you.

  1. What specific improvements should the authors consider regarding the
    methodology? 

The evaluation of “itch behavior” using ‘yes/no’ could be improved. Understandably, the quantitative evaluation is very time-consuming, but should not be overlooked. Maybe a simplified quantitative evaluation can be considered in the future.    

Answer to the comment: Thank you for the comments. We pointed out the issues of video evaluation for specific pruritic behaviors in the discussion part below. Our goal was a quantifiable method but we could not do that for a reason below:

In the Discussion part of the manuscript:

“Among different pruritic behaviors, scratching, head shaking and biting/chewing were most commonly recorded utilizing the categorical “YES/NO” per 1 min interval scoring after the intradermal injection of canine IL-31 in our study. Our initial intention was to assess every different itch behavior in our IL-31 study and quantify the duration in seconds. The initial investigations of video recordings showed that analyzing all pruritic behaviors in total seconds can be easily performed; however, the analysis of consecutive changes in different pruritic behaviors in seconds utilizing the slow-forward function on video recordings increased the duration of video evaluations above 10 hours per dog. Although our categorical “YES/NO” per discrete 1 min scoring system for different pruritic behaviors may contain a bias and is a limitation of our study, the authors felt that the canine IL-31 pruritic model evaluations need to be performed in a time-efficient way.”

We agree that we would like to have a better quantifiable way of measuring every behavior. However, in our study, we found it very hard to evaluate specific itch behaviors in seconds by separating these since dogs showed continuous mix (change from head shaking to licking to head shaking to chewing, then rubbing etc) of pruritic behaviors after IL-31 administration.

We are working now with motion sensors and we hope these work since they the motion sensors can be quantified. The main limitation is that the motion sensors don’t record all the pruritic behaviors.

What further controls should be considered? 

Using different dog breed should be considered. 
Answer to the comment: We agree with you. However, research Beagles are the most commonly utilized canine research breed and many studies, including house dust mite-sensitized allergic canine models, were performed on Beagles. There is a limitation, of course, to this, but the model serves only as a model, basically proof of concept.

  1. Are the conclusions consistent with the evidence and arguments presented
    and do they address the main question posed? 

Yes. This study is quite straight forward and the conclusions addressed the main
Answer to the comment: Thank you

  1. Are the references appropriate? Yes.

    Answer to the comment: Thank you
  2. Please include any additional comments on the tables and figures. None.

Answer to the comment: Thank you

Reviewer 2 Report

Typo on last sentence of Introduction: [IL-31-indcued canine pruritic] instead of [IL-31-induced canine pruritic]

1. What is the main question addressed by the research? Yes

2. Do you consider the topic original or relevant in the field? Does it

address a specific gap in the field? Yes/Yes. 

3. What does it add to the subject area compared with other published

material? Canine preclinical model for anti-allergic/itch drugs

4. What specific improvements should the authors consider regarding the

methodology? What further controls should be considered? I think this paper address well methodology 

5. Are the conclusions consistent with the evidence and arguments presented

and do they address the main question posed? Yes

6. Are the references appropriate? Yes

7. Please include any additional comments on the tables and figures. Probably, it'd be nice to have a figure displaying the pruritic changes (PVA scale) vs time and groups (vehicle vs IL-31) and either embedded or as supplement.

Author Response

Typo on last sentence of Introduction: [IL-31-indcued canine pruritic] instead of [IL-31-induced canine pruritic]

Answer to the comment: Thank you, we corrected that.

  1. What is the main question addressed by the research? Yes

Answer to the comment: Thank you

  1. Do you consider the topic original or relevant in the field? Does it

address a specific gap in the field? Yes/Yes. 

Answer to the comment: Thank you

  1. What does it add to the subject area compared with other published

material? Canine preclinical model for anti-allergic/itch drugs
Answer to the comment: Thank you

  1. What specific improvements should the authors consider regarding the

methodology? What further controls should be considered? I think this paper address well methodology 

Answer to the comment: Thank you

  1. Are the conclusions consistent with the evidence and arguments presented

and do they address the main question posed? Yes

Answer to the comment: Thank you

  1. Are the references appropriate? Yes

Answer to the comment: Thank you

  1. Please include any additional comments on the tables and figures. Probably, it'd be nice to have a figure displaying the pruritic changes (PVA scale) vs time and groups (vehicle vs IL-31) and either embedded or as supplement.
    Answer to the comment: Thank you for the comment. We are unsure what the reviewer exactly understands under the PVA scale.

We presume PVA sclai is the pruritus visual analogue scale, freuqnetly utilized in clinical studies evaluating itch by owners

In the past, there have been several studies evaluating canine itch behaviors where the investigator would sit in front of the dog and try to evaluate itch. These contain a basic flow that the dogs will change their behaviors and activity in the presence of humans in front of their cage. In our study, we cannot use PVA scale since our presence in the room would distract dogs and they will modify their behaviors. That entails a significant bias and cant be published (although these studies were published back then). We used video recordings as stated in the materials and methods.  

Reviewer 3 Report

This is the first paper to examine the pruritogenic potential of IL-31 injected locally in dogs. The experimental design is sound, and the findings novel - in that a delayed onset of action was found. The pruritogenic effects were prevented by prior treatment with oclacitinib. 

The experimental approach is in general well-presented, however this reviewer found one aspect confusing. In the study recording the different type of pruritic behaviors, it is stated that a categorical scoring system "yes/no" (i.e. 1 or 0) at one minute intervals was determined. However in fig 3 the data is presented as "yes/no" per minute. Does this imply that multiple observations were made within a one minute period, or that at every minute one "yes/no" observation was recorded?

Author Response

This is the first paper to examine the pruritogenic potential of IL-31 injected locally in dogs. The experimental design is sound, and the findings novel - in that a delayed onset of action was found. The pruritogenic effects were prevented by prior treatment with oclacitinib.

Answer to the comment: Thank you.

The experimental approach is in general well-presented, however this reviewer found one aspect confusing.

In the study recording the different type of pruritic behaviors, it is stated that a categorical scoring system "yes/no" (i.e. 1 or 0) at one minute intervals was determined. However in fig 3 the data is presented as "yes/no" per minute. Does this imply that multiple observations were made within a one minute period, or that at every minute one "yes/no" observation was recorded?

Answer to the comment: Thank you for the comments. We apologize for any confusion.

The Figure 3 shows data for each pruritic behavior.

In the Materials and Methods, we state that we had to use YES/NO categorical scoring for each pruritic behaviors, although our initial desire was to use pruritic seconds.

Here is the text from Materials and Methods:

“The video evaluators initially tried to quantify each type of pruritic behavior (e.g., licking, chewing, head shaking) in total seconds; however, repeated videos evaluations in slow motion needed to be performed, which resulted in an extensive time commitment for every video evaluation. To simplify the assessment of types of pruritic behaviors and make it time-efficient for future study evaluations, observed pruritic behavior (i.e., licking, biting/chewing, scratching, head shaking, scooting, rolling, pawing and rubbing) and sleeping was measured using a categorical scoring system ‘Yes ⁄ No’ (i.e., 1 or 0) at 1 min intervals.”

We follow up on this issue in the Discussion as well. Here the text below:  

In the Discussion part of the manuscript:

“Among different pruritic behaviors, scratching, head shaking and biting/chewing were most commonly recorded utilizing the categorical “YES/NO” per 1 min interval scoring after the intradermal injection of canine IL-31 in our study. Our initial intention was to assess every different itch behavior in our IL-31 study and quantify the duration in seconds. The initial investigations of video recordings showed that analyzing all pruritic behaviors in total seconds can be easily performed; however, the analysis of consecutive changes in different pruritic behaviors in seconds utilizing the slow-forward function on video recordings increased the duration of video evaluations above 10 hours per dog. Although our categorical “YES/NO” per discrete 1 min scoring system for different pruritic behaviors may contain a bias and is a limitation of our study, the authors felt that the canine IL-31 pruritic model evaluations need to be performed in a time-efficient way.”

We agree that we would like a better quantifiable way of measuring every behavior. However, in our study we found it very hard to evaluate specific itch behaviors in seconds by separating these since dogs showed continuous mix (change from head shaking to licking to head shaking to chewing, then rubbing etc) of pruritic behaviors after IL-31 administration.

We are working now with motion sensors and hope these works since they can be quantified. The main limitation is that the motion sensors don’t record all the pruritic behaviors.
